# Gone or just out of sight? The apparent disappearance of aromatic litter components in soils

**T. Klotzbücher[1], K. Kalbitz[2], C. Cerli[3], P.J. Hernes[4], K. Kaiser[1]**

[1] Soil Science and Soil Protection, Martin Luther University Halle-Wittenberg, von-Seckendoff-Platz 3, 06120 Halle (Saale), Germany

[2] Institute of Soil Science and Site Ecology, Technical University Dresden, Pienner Strasse 19, 01737 Tharandt, Germany

[3] Institute of Biodiversity and Ecosystem Dynamics, Earth Surface Science, University of Amsterdam, POSTBUS 94240, 1090 GE Amsterdam, The Netherlands

[4] Department of Land, Air, and Water Resources, University of California, One Shields Avenue, Davis, California 95616, United States

Correspondence to: T. Klotzbücher (thimo.klotzbuecher@landw.uni-halle.de)

## Abstract

Uncertainties concerning stabilization of organic compounds in soil limit our basic understanding on soil organic matter (SOM) formation and our ability to model and manage effects of global change on SOM stocks. One controversially debated aspect is the contribution of aromatic litter components, such as lignin and tannins, to stable SOM forms. In the present opinion paper, we summarize and discuss the inconsistencies and propose research options to clear them.

Lignin degradation takes place step-wise, starting with (i) depolymerisation, followed by (ii) transformation of the water-soluble depolymerization products. The long-term fate of the depolymerization products and other soluble aromatics, e.g., tannins, in the mineral soils is still a mystery. Research on dissolved organic matter (DOM) composition and fluxes indicates dissolved aromatics are important precursors of stable SOM attached to mineral surfaces and persist in soils for centuries to millennia. Evidence comes from flux analyses in soil profiles, biodegradation assays, and sorption experiments. In contrast, studies on composition of mineral-associated SOM indicate the prevalence of non-aromatic microbial-

derived compounds. Other studies suggest the turnover of lignin in soil can be faster than the turnover of bulk SOM. Mechanisms that can explain the apparent fast disappearance of lignin in mineral soils are, however, not yet identified.

The contradictions might be explained by analytical problems. Commonly used methods probably detect only a fraction of the aromatics stored in the mineral soil. Careful data interpretation, critical assessment of analytical limitations, and combined studies on DOM and solid-phase SOM could thus be ways to unveil the issues.

## 1   Introduction

Storage and quality of soil organic matter (SOM) determine many crucial soil properties and the cycling of carbon (C) and essential nutrients through ecosystems. The storage of SOM is determined by plant litter inputs and decomposition processes. Decomposition of SOM is a significant source of atmospheric $CO_2$, thus, a critical parameter in climate models (Schlesinger and Andrews, 2000). Decomposition rates are sensitive to global change factors such as temperature, precipitation, and land use. However, our ability to understand and predict such responses is limited by uncertainties about pathways of organic matter transformation in soil. In particular, the question as to why some SOM components persist in soil for centuries (denoted as `stable SOM` from here on) while others turn over quickly is still puzzling (Schmidt et al., 2011).

Recent research challenges traditional theories presuming that stable SOM results from neoformation of complex humic polymers in soil (`humification`). Stable SOM rather seems to be composed of relatively simple organic compounds that are protected against biodegradation, e.g., because they are tightly bound to mineral surfaces (Schmidt et al., 2011; Kleber et al., 2015). Herein, we hold to this view but argue that, despite extensive research in the last years, the chemistry and source of compounds incorporated into stable SOM is still largely uncertain. In particular, the importance of aromatic compounds derived from abundant plant litter components, such as lignin and tannins, is controversially debated (Figure 1). One line of evidence suggests that they are important contributors to stable SOM. It bases primarily on data from research on fluxes and behaviour of dissolved organic matter (DOM) in soil, hence, we will denote it as the `dissolved phase line of evidence`. A contrasting line of evidence suggests a quick degradation of aromatic compounds in soil derives primarily from analyses of the composition of solid SOM (`solid phase line of evidence`). Herein, we sum up

and confront the arguments of the two views, discuss potential reasons for the controversies (including limitations in analytical methods and process understanding) as well as their implications for our basic understanding of SOM formation.

## 2   Dissolved phase line of evidence

The view that plant-derived aromatics are a major source of stable SOM is based on the following main arguments:

(1) DOM produced during litter decomposition and leached into mineral soil is a main source of stable SOM adsorbed on mineral surfaces.

(2) Aromatic DOM components produced during litter decomposition are resistant to mineralization and preferentially sorb to mineral surfaces. Hence, they are preferentially stabilized in mineral soil.

(3) Root decomposition in mineral soil could be another important source of aromatic DOM components that contribute to stable SOM.

### 2.1   Argument 1: DOM as source of stable SOM

Leaching of DOM is a major pathway for organic matter translocation from forest floor into the topsoil horizons. Estimates for acidic forest soils with permanent forest floor suggest that 25-89% of the SOM stored in mineral soils derives from DOM (Neff and Asner, 2001; Michalzik et al., 2003; Kalbitz and Kaiser, 2008), based on the typical observation of decreasing DOC fluxes with depth of the mineral soil (a large compilation of data from studies on forest and grassland soils is presented by Neff and Asner, 2001). Two processes can explain the decrease: mineralization and sorption.

Sorption of DOM to mineral surfaces likely is a major process forming stable SOM in many soils. Evidence for its importance comes from findings that the turnover and storage of SOM in mineral soil horizons is related to the contents of reactive secondary minerals (e.g., Fe hydrous oxides, short-range ordered Al hydroxides). Such relationships have been found across a wide range of soil types (comprehensive data sets have been presented in Torn et al., 1997, Eusterhues et al., 2005, Kögel-Knabner et al., 2008, Kramer et al., 2012, and Kleber et

al., 2015). Also, higher radiocarbon age of SOM in heavy (i.e., mineral) fractions compared to light density (i.e. organic) fractions indicates that sorption stabilizes organic compounds (see data compilations by Marschner et al., 2008, Kögel-Knabner et al., 2008, and Kleber et al., 2015). Density fractionation procedures indicate that the total soil C associated with minerals in any given location can vary from 30% to 90% (data compiled in Kleber et al., 2015). The relevance of sorptive stabilization depends on soil properties. Low soil pH enhances the formation of reactive secondary minerals and favors the formation of strong bonds between organic matter and the mineral surface (Kleber et al., 2015). Most studies cited herein (for both lines of evidence) examined acidic soils under temperate forests, in which sorptive stabilization clearly should play an important role for the long-term storage of organic matter in soil.

## 2.2   Argument 2: Preferential stabilization of aromatic DOM components

Lignin, a macromolecule composed of phenyl propane units, is a major plant cell wall component (Kögel-Knabner, 2002). Typically, lignin concentrations negatively correlate with litter decomposition rates. They are the predominant control on litter decomposition within biomes worldwide (Cornwell et al., 2008), indicating that the lignin macromolecule is among the most persistent litter constituents. Nevertheless, results of recent studies suggest significant chemical alteration and losses of lignin already within the first months and years of litter decomposition (Kalbitz et al., 2006; Preston et al., 2009; Klotzbücher et al., 2011; Duboc et al., 2014). 'Degradation' of lignin has to be considered a step-wise process: (i) the first step is the depolymerization of the macromolecule, releasing (mainly aromatic) water-soluble depolymerisation products of varying molecular weight; (ii) these products can then be further transformed, and low-molecular weight compounds are eventually taken up by microorganisms to produce biomass or $CO_2$. Hence, losses of lignin-derived C during litter decomposition can occur due to leaching of water-soluble products of an incomplete degradation or as $CO_2$. Laboratory incubation tests on water-extractable organic matter from various plant and soil materials suggest that aromatic components are more resistant to mineralization than non-aromatic components (Pinney et al., 2000; Kalbitz et al., 2003 a,b; Marschner and Kalbitz, 2003; Don and Kalbitz, 2005; McDowell et al., 2006; Hagedorn and Machwitz, 2007; Fellman et al., 2008; Hansson et al., 2010; Kothawala et al., 2012; Toosi et al., 2012). This suggests that leaching is an important factor in loss of lignin-derived matter

during litter decomposition. Consistent with this conceptual model, the typically high UV absorptivity of DOM leached from forest floors (i.e., higher than values found for DOM leached from Oi and Oe horizons) is indicative of a large contribution of resistant aromatic components (Kalbitz et al., 2007; Fröberg et al., 2007).

Another factor for the export of aromatic DOM from forest floors is leaching of tannins. Tannins are water-soluble polyphenols of a molecular weight ranging from 500 to 3000 Daltons. Tannins rapidly leach from fresh litter; most studies suggest losses of ~80% within the first year of litter decomposition (Kraus et al., 2003).

It has been commonly found that the contribution of components likely derived from lignin and tannins to DOM decreases with depth of the mineral soil (summarized in Table 1), i.e., the decrease in fluxes of these compounds with depth is more pronounced than the decrease of bulk DOM. One explanation might be intensive biodegradation of aromatics in mineral soil. However, this would contradict results of the DOM biodegradation studies previously discussed. Hence, a more likely explanation is sorption to mineral surfaces. Laboratory sorption experiments support this view; a typical observation is that aromatic DOM components are preferentially sorbed by minerals and soils (Davis and Gloor, 1981; Jardine et al., 1989; McKnight et al., 1992; Dai et al., 1996; Kaiser and Zech, 1997, 2000; Chorover and Amistadi, 2001; Guo and Chorover, 2003; Kalbitz et al., 2005; Kawahigashi et al., 2006; Hunt et al., 2007; Mikutta et al., 2007; Jagadamma et al., 2012; Sodano et al., 2016), and for some soils it has been shown they displace previously bound organic components from mineral surfaces (Kaiser et al., 1996). The degree of preferential sorption may depend on the composition of the soil mineral assemblage. Chorover and Amistadi (2001) observed that high molecular weight aromatic components preferentially sorbed onto goethite, while for montmorillonite no preference for aromatic moieties was observed. A likely reason for the preferential sorption is the large content of carboxyl groups linked to the aromatic rings, which bind to metals at mineral surfaces via ligand exchange reactions.

## 2.3. Argument 3: Roots as source of stabilized aromatic SOM

We have so far focused on DOM leached from aboveground litter. However, roots might also be a crucial source of stable SOM. The contribution of root and aboveground litter as major source of SOM has been debated in numerous studies, but the available information allows no definite conclusions yet (comprehensive discussions on the topic can be found in Lajtha et al.,

2014, and Hatton et al., 2015). Presumably, the relative importance of the two types of organic matter input for SOM storage in topsoils differs between ecosystems (Crow et al., 2009) and the importance of root-derived matter increases with soil profile depth (Rumpel et al., 2015).

This raises issue of whether results from aboveground litter decomposition would also apply to root litter decomposition. Data by Crow et al. (2009) suggest that lignin concentrations of roots are in the range of those of leaf and needle litter. Root-derived DOM shows higher concentrations of aromatic compounds than DOM from foliar litter (Hansson et al., 2010; Uselman et al., 2012). Hansson et al. (2010) showed that DOM production during root decomposition occurs in patterns that are similar to those of needle decomposition. Particularly during later decomposition stages, root-derived DOM is enriched in aromatics resistant to mineralization. Hence, available information suggests that root decomposition is just another important source of soluble aromatics in mineral soils. However, to the best of our knowledge, studies to quantify the contribution of root-derived aromatics to DOM fluxes in the field have yet to be conducted. Overall, the available information is limited to data from laboratory experiments and refers to acid temperate forest soils, so that it is not yet possible to draw general conclusions.

## 3   Solid phase line of evidence

Many of the recent conceptual papers on SOM formation are built on the assumption that lignin-derived aromatics disappear quickly in soil, while SOM in mineral soils is dominated by non-aromatic and microbial-derived compounds (Grandy and Neff, 2008; Schmidt et al., 2011; Dungait et al., 2012; Miltner et al., 2012; Cotrufo et al., 2013; Castellano et al., 2015). Empirical support is provided by studies characterizing the chemical structure of solid SOM using a variety of analytical methods.

Numerous studies on a wide variety of soil types used the cupric oxide (CuO) method to analyse the distribution of lignin-derived phenols in profiles. A typical observation is that the phenol contribution to SOM decreases (i) from forest floor to A horizons, (ii) with depth of the mineral soil, (iii) with decreasing soil particle size (reviewed in Thevenot et al., 2010) and with increasing density of soil fractions, hence, are contributing little to heavy (i.e., mineral-associated) and old soil fractions (Leifeld and Kögel-Knabner, 2005; Grünewald et al., 2006; Sollins et al., 2009; Kögel-Knabner et al., 2008; Cerli et al., 2012). Similar results are

reported by studies using pyrolysis-gas chromatography/mass spectrometry (Gleixner et al. 2002; Nierop et al., 2005; Buurman et al., 2007; Grandy and Neff, 2008; Tonneijck et al., 2010: Rumpel et al., 2012) and tetramethylammonium hydroxide (TMAH) thermochemolysis (Nierop and Filley, 2007; Mason et al., 2012).

Analysis of heavy and clay-sized soil fractions using cross polarization and magic angle spinning (CPMAS) $^{13}$C nuclear magnetic resonance spectroscopy ($^{13}$C-NMR) typically finds high peak intensities of alkyl and O/N alkyl C (mostly assigned to polysaccharides and proteins) and low peak intensities of aryl C (mostly assigned to lignin and tannins) (see data compilations by Mahieu et al., 1999, and Kögel-Knabner et al., 2008). For instance, in a comprehensive study on Ah horizons from 8 European forest sites, O/N alkyl C contributed up to 41-49% of total peak intensity in the <2-µm fraction, and the peak intensities were on average 10% higher than the those reported for bulk soil; the intensities of aryl C in the <2-µm fraction contributed 13-15% of total peak intensities, and they were on average 24% lower than values found for bulk soil (Schöning et al., 2005). Also studies using near-edge X-ray absorption fine structure (NEXAFS) spectra supported the conclusions drawn from $^{13}$C-NMR analysis of a significant contribution of microbial-derived compounds to SOM at mineral surfaces (Lehmann et al., 2007; Kleber et al., 2011).

Consistent with these findings, the heavy soil fraction is characterized by low C/N values close to those of microbial tissues (Kögel-Knabner et al., 2008).

Analysis of microbial-derived polysaccharides by acid hydrolysis suggest an enrichment of these compounds in fine and heavy soil fractions (Kiem and Kögel-Knabner, 2003; Rumpel et al., 2010).

Lignin turnover (i.e., transformation into $CO_2$ or non-lignin products) in temperate arable, grassland, and tropical forest soils has been estimated using a combination of isotopic labeling and compound-specific isotope analysis of lignin-derived aromatics applying the CuO method. Most of the studies using this approach suggest that the turnover of lignin-derived aromatics is faster than the turnover of bulk SOM (Dignac et al., 2005; Heim and Schmidt, 2007; Heim et al., 2010). A modeling study based on the data by Dignac et al. (2005) suggested that about 90% of the lignin is mineralized as $CO_2$ or transformed into compounds devoid of lignin-type signatures within one year (Rasse et al., 2006). However, a study by Hofmann et al. (2009) suggests that after 18 years, approximately two-thirds of the initial

lignin phenols remained in an arable soil. The authors concluded that lignin was preferentially preserved in the soil.

## 4    Reasons for the controversies

### 4.1.    Analytical limitations

The controversies in current literature might (partly) be due to difficulties in the analyses of aromatic OM compounds in soils. Studies on DOM typically use bulk methods for inferring aromatic content, including UV absorbance and fluorescence spectroscopy. Limitations of this research include lack of identification of the source of aromatic compounds, and poor quantification of the fluxes. Also data on contribution of aromatic components to solid SOM are semi-quantitative or qualitative.

Commonly applied methods such as CuO oxidation, pyrolysis or TMAH thermochemolysis focus on few defined lignin-derived monomers to estimate the overall contribution of lignin. These estimates, however, can largely differ depending on the method applied (Klotzbücher et al., 2011). As outlined by Amelung et al. (2008), compound-specific isotope analysis of lignin-derived compounds with the CuO method presumably overestimates the turnover rates of lignin as only part of the lignin-derived aromatics can be extracted from soil (incomplete extraction might also be a problem in all analyses of biomarkers, for which turnover times typically are estimated to be faster than turnover rates of bulk SOM). Firstly, CuO oxidation (as well as conventional pyrolysis or TMAH thermochemolysis) does not completely depolymerize lignin (Johansson et al., 1986; Goňi and Hedges, 1992; Filley et al., 2000). Secondly, lignin-derived aromatics bound to mineral surfaces are only partly assessed by the CuO method (Hernes et al., 2013). Thirdly, lignin-derived aromatics might be altered in a way that they escape the `analytical window` and cannot be ascribed to a lignin source anymore. For instance, the CuO method yields a number of aromatic monomers of unknown origin besides the lignin-derived monomers (Cerli et al., 2008). These compounds are typically not quantified, and thus, not considered in estimates of the SOM composition. Hence, monomer yield is a commonly used but uncertain measure of lignin concentration in soil.

Solid-state CPMAS $^{13}$C-NMR has been widely used in the last decades to study the composition of SOM. Whether the results are quantitative has been subject of an intensive

debate (see Knicker, 2011). Mineral soil samples are commonly pretreated with hydrofluoric acid (HF) in order to remove paramagnetic minerals that disturb the analysis. The treatment can result in significant losses of SOM, and one might lose important information on SOM adsorbed onto minerals (e.g., SOM losses of 10-30% in topsoil samples and up to 90% in subsoil; Eusterhues et al., 2003). Eusterhues et al. (2007) attempted to assess the chemical composition of HF-soluble SOM by comparing CPMAS [13]C-NMR spectra of untreated and HF-treated soil samples. The data suggest that the composition of HF-soluble SOM varies between soil type and soil horizons. The effect of HF treatment, thus, produces unpredictable changes in composition and questions the meaning of spectra obtained on HF-treated samples. However, the approach used comes with the uncertainty that in the untreated soil part of the SOM attached to mineral surfaces might have been invisible due to the proximity to paramagnetic material (Kinchesh et al. 1995), while in the HF-treated soil, the same SOM might have become removed during the treatment, thus, was no more detectable. Further uncertainties of CPMAS [13]C-NMR arise from signal overlapping and a general low sensitivity for aromatic C in soils (e.g., Skjemstad et al., 1996; Mao et al., 2000; Simpson and Simpson, 2012).The common approach used to quantify the relative contribution of different C types to SOM is to integrate and compare peak areas of different spectra regions without considering any non-proportional signal responses (e.g., Kögel-Knabner 2002). It has been shown that this approach underestimates lignin vs. cellulose in ligno-cellulose isolated from wheat (Gauthier et al., 2002). Also, peaks of methoxyl C often tend to be larger than those of aromatic C. Both, methoxyl C and aromatic C derive from lignin, but lignin has much less methoxyl C than aromatic C. Spectra of wood samples (e.g., Bonanomi et al. 2014), needle litter (Preston and Trofymow, 2015) and grass litter (McKee et al., 2016) all show that patterns of methoxyl signals being larger than those of aromatic and phenolic C combined.

By applying Bloch decay, another type of [13]C-NMR technique, one can overcome the problem of the reduced sensitivity for aromatic C. The technique has been applied in studies on pyrogenic organic matter, for which CPMAS [13]C-NMR should be even less sensitive than for lignin as it is more condensed (Golchin et al., 1997; Simpson and Hatcher, 2004; Knicker et al., 2005). Bloch decay, however, also comes with problems, such as general low signal intensity. It is not routinely applied in SOM research as it is an extremely time-consuming experiment and the required instrument time is frequently not available (Simpson and Hatcher, 2004).

## 4.2.   Limits in process understanding

The contradictions outlined herein might also suggest gaps in the understanding of SOM turnover processes. Here we argue that, in particular, knowledge about the turnover of SOM at mineral surfaces is insufficient. This is due to the yet uncertain quantitative composition of SOM. In addition, prevailing conceptual ideas and paradigms have been questioned in recent years.

It has been frequently observed that the $^{14}C$ age of DOM increases with profile depth. Kaiser and Kalbitz (2012) proposed that this can be explained by temporal sorptive immobilization, followed by microbial processing and re-release of altered compounds into soil solution. That would mean the assumption that aromatic compounds are stable after being sorbed onto mineral surfaces could be erroneous. Hence, the microbial processing of sorbed compounds might be the `missing argument` that proves the view of a fast disappearance of aromatic compounds in mineral soil. However, these processes have hardly been studied yet, and empirical evidence for their importance is missing.

Also, root activity might have significant effects on stability and composition of SOM sorbed onto mineral surfaces. The recent study of Keiluweit et al. (2015) showed that root exudation of oxalic acid promotes release of sorbed compounds into soil solution through dissolution of mineral phases. Besides oxalic acid, many other low-molecular weight compounds, including acids, simple sugars, amino sugars, phenolics, as well as high-molecular compounds (exoenzymes, root cells) are released from living roots (Wichern et al., 2008). Most of the compounds are degraded quickly, but a smaller portion seems to contribute to stable SOM (Nguyen, 2003; Pausch et al., 2013). The rhizosphere is considered a `hot-spot` in soil, where microbial processes are not C-limited (Kuzyakov and Blagodatskaya, 2015). Hence, root activity possibly may accelerate the turnover of sorbed plant-derived aromatics, and at the same time foster the production and stabilization of microbial-derived compounds. Studies addressing these assumptions are not yet available.

Recent investigation at the submicrometer scale using Nano Secondary Ion Mass Spectrometry (NanoSIMS) or NEXAFS in combination with scanning transmission X-ray microscopy suggest that SOM associated with clay-sized minerals exists in small patches of varying chemical composition (Lehmann et al., 2008; Remusat et al., 2012; Vogel et al., 2014). Distinct patches of predominantly aromatic C can be differentiated from patches

dominated by aliphatic C (Lehmann et al., 2008). Knowledge about processes controlling the sub micro meter-scale distribution of SOM on mineral surfaces is still limited. Some of the patches are cell wall structures of microorganisms, which may contribute to stable SOM as they are composed of insoluble polymers and possibly attach to the mineral surface (Miltner et al., 2012). Hence, stable sorbed organic matter might not only be derived from low-molecular weight compounds. On the other hand, it needs to be considered that microbial-derived compounds are continuously synthesized at the mineral surface. The microorganisms might use some of the older C ($^{14}$C age) for synthesis of relatively labile compounds. The age of the C atoms is, thus, decoupled from the stability of the organic matter, and microbial-derived compounds may `mimic` a similar or even higher stability than the plant-derived compounds (Gleixner, 2013). Hence, concentrations or $^{14}$C age of microbial-derived compounds at mineral surfaces do not per se allow for conclusions on their contribution to stable SOM.

Pyrogenic organic matter is an important source of aromatic compounds in many soils. Despite extensive research efforts, rates, and pathways of pyrogenic organic matter decomposition are still not well established (Schmidt et al., 2011; Kuzyakov et al., 2014). Analyses of benzenecarboxylic acids as molecular markers suggest that aromatic compounds derived from pyrogenic organic matter are transported within soil profiles and bind to mineral surfaces (Haumaier, 2010). The quantitative contribution of pyrogenic organic matter to DOM in soil is, however, still poorly studied (Smebye et al., 2016). Bulk analyses of aromatic matter used in most research on DOM fluxes (i.e., UV absorption) cannot distinguish if the compounds derive from plant litter or from pyrogenic organic matter. This limits the understanding of the processes controlling turnover of aromatics. If a significant part of aromatic DOM in mineral soil derives from pyrogenic organic matter, the `loss` of plant litter-derived aromatics in mineral soil would be even more pronounced.

## 5   Implications and future research strategies

The contradictions outlined herein limit our basic understanding on SOM formation, and our ability to model and manage effects of global change on SOM stocks.

For instance, elevated atmospheric $CO_2$ levels can induce increasing concentrations of aromatic components in plant litter (Cotrufo et al., 1994; Tuchman et al., 2002), and this raises the question whether this causes enhanced or decreasing storage of SOM in mineral

soils. If aromatic matter is quickly degraded, and mineral-associated SOM primarily derives from microbial sources (as suggested by the solid SOM line of evidence), a `microbial filter` would control the built-up of stable SOM, which may then be determined by the microbial substrate use efficiency (i.e., the amount of organic C used by the microbial community to build biomass vs. the amount that is mineralized). As such, Cotrufo et al. (2013) hypothesized that input of labile substrates fosters the build-up of stable SOM. Available data on effects of litter quality and SOM formation are, however, inconsistent (Castellano et al., 2015), and we believe that understanding these effects is in part limited by uncertainties about the incorporation of aromatics into stable SOM.

The issue is also related to the question of links between chemical structure of organic matter and its persistence. It is oftentimes assumed that structural properties of plant-derived matter do not determine stable SOM formation. This argument is based on data suggesting that specific compound classes (lignin, cellulose, alkanes, proteins etc.) turn over faster than bulk SOM (Schmidt et al., 2011). However, conclusions of DOM research imply that structure plays a role for the behavior of organic compounds in soil, and eventually their contribution to stable SOM: soluble aromatics may resist oxidation by microbes as they yield less energy than other structures; furthermore, they bind to mineral surfaces due to carboxyl groups attached to the rings.

How could we resolve the controversies? Based on our literature analysis we propose the following research strategies:

- Knowledge on aromatics in soils is limited by the analytical constraints. Quantification of total amounts and sources of aromatics in soil are still problematic. Even if the problems cannot be fully solved with the currently available techniques, there might be strategies to obtain improved estimates. The work of Hernes et al. (2013) provides a first hint about how much lignin might be not accessible to CuO oxidation analyses. The authors evaluated the extraction efficiency for lignin-derived aromatics bound to different minerals. The size of the non-extractable fraction depended on the mineral. Almost all of the aromatics bound to ferrihydrite were extractable, but for kaolinite the non-extractable fraction made >40%. But how about extractability in soil under field conditions? Possibly, a combination of tracking of C isotopes, DOM flux/composition assessment, and analysis of solid-phase SOM composition could provide better estimates of hidden aromatics.

- The causes of the commonly observed decreasing fluxes of aromatic DOM fluxes with depth of the mineral soil need to be re-examined. Are they really mainly the result of sorption to mineral surfaces (as proposed herein in Chapter 2), or do other processes such as the binding of tannins to proteins or mineralization also play a decisive role? Moreover, the presumed microbial processing of sorbed material, causing desorption and subsequent mineralization or further transport in the soil profile is poorly studied. Knowledge gaps also exist concerning the question whether root activity affects de-/sorption processes. Eventually, these processes might cause loss of aromatic compounds.

- Computer simulations could help to unravel the complex interrelationships between DOM fluxes and solid-phase SOM composition. Recently developed models integrate sorption, DOM transport, and microbial processes (Ahrens et al., 2015). In order to address the problems discussed herein, effects of molecular structure on behavior of the compounds in soil (e.g., differences in mineralization rate and affinity for sorption between aromatics and non-aromatics) could be implemented in the models, in order to develop novel hypotheses on turnover of plant-derived aromatics.

**Acknowledgements**

TK acknowledges the support of his research on lignin degradation from the German Research Foundation (DFG).

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

28

1 Table 1. Evidence from field studies suggesting that dissolved aromatics (products of lignin

2 depolymerization or tannins) disappear quickly once entering mineral soils.

| Reference | Study site/ soil type | Result |
| --- | --- | --- |
| Qualls and Haines 1991 | Oak-hickory forest in mountain region of North Carolina; soil types: Umbric Dystrochrept, Typic Hapludult, Typic Dystrochrept | Selective removal of hydrophobic acids as DOM percolates through the mineral soil. |
| Cronan 1985 | Forests, North-Western USA; soil types: Dystrochrept, Haplorthod | Selective removal of hydrophobic acids as DOM percolates through the mineral soil. |
| Zech et al. 1994 | Spruce forest in Bavaria, Germany; Soil types: Typic Dystrochrepts, Entic Haplorthods, Typic Haplorthods | Selective removal of lignin-derived phenols (determined with the CuO method) as DOM percolates through the mineral soil. |
| Gallet and Pellissier 1997 | Bilberry-spruce forest in Alps, France; soil type: Humoferric Podzol | Selective removal of lignin-derived phenols (as well as of total phenols) as DOM percolates through the mineral soil. |
| Kaiser et al. 2004 | Spruce forest in Bavaria, Germany; Soil type: Haplic Arenosol | Selective removal of lignin-derived phenols, hydrophobic compounds, and total aromatic C as DOM percolates through the mineral soil. |
| Dai et al. 1996 | Spruce forest, Maine; Soil type: Aquic Haplothods | Selective removal of hydrophobic acids and aromatics ($^{13}$C-NMR data) as DOM percolates through the mineral soil. |
| Lajtha et al. 2005 | Coniferous forest, Oregon, USA; soil type: Typic Hapludands | Selective removal of hydrophobic acids as DOM percolates through the mineral soil. |
| Sanderman et al. 2008 | Mediterranean climate; forest and grassland soils; soil types: Haplustols and Haplohumults | Decrease in UV absorbance (a measure for content of aromatics) as DOM percolates through the mineral soil. |
| Hassouna et al. 2010 | Mediterranean climate; maize field; soil type: fluvic hypercalcaric cambisol | Decrease in contents of aromatic compounds (UV absorbance, fluorescence specroscopy) in water-extractable organic matter with depth of the mineral soil. |
| Nakashini et al. 2012 | Beech forest, Japan; soil: "brown forest soil" | Decrease in contents of hydrophobic acids in water-extractable organic matter with depth of the mineral soil. |

1    Figure 1. Conflicting views on the fate of soluble aromatics once they enter the mineral soil

2    (see text for references).

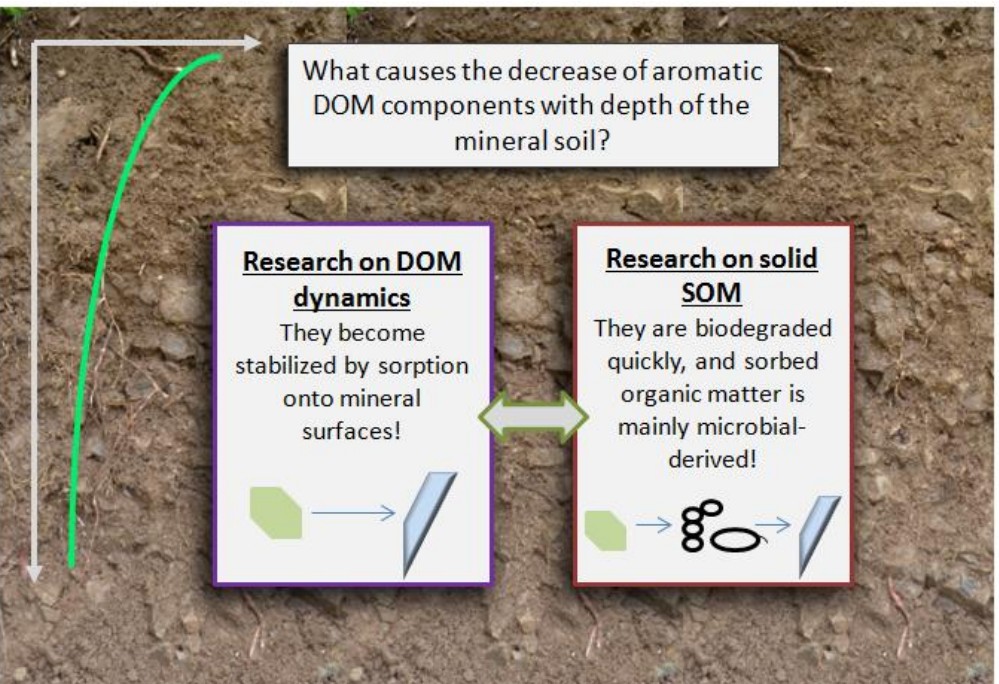

