# Peer review of "Gone or just out of sight? The apparent disappearance of"

_SOIL, 2015_

## Referee Comment (RC1) · Anonymous Referee #1 · 10 Feb 2016

I generally acknowledge that the authors try to pinpoint and discuss the apparent discrepancies between the fact that large amounts of aromatic compounds are entering the soil in dissolved phase and the fact that they are not found any more sorbed to the solid phase. The authors attribute this to the problems of analyzing lignin and claim that two methods (although completely independent and different in their analytical procedures, i.e. solid-state 13C NMR spectroscopy and CuO oxidation) fail to identify these aromatic compounds. I suggest to carefully check the literature for solid-state 13C NMR work that shows significant contributions of aromatic compounds (although mostly attributed to charred OM, see work by Knicker and coworkers, Skjemstad and coworkers). The authors need to explain why the technique fails to work for lignin-derived aromatic compounds, but does work well for other aromatic (and even more condensed) structures produced from fire impact. The problem that all compounds en-

tering the soil after some decades leave the analytical window for and cannot be identified any more as specific plant or microbial derived compounds has been described and discussed previously and is not specific for aromatic compounds (see detailed discussion of the problem in Hedges et al. (2000). It is also illustrated by the fact that all molecularly identified organic compounds in soil are younger than the mean age of SOM or their turnover is faster than that of bulk SOM (see Amelung et al., Adv in Agronomy, 2008 and later Schmidt et al., Nature, 2011). Thus I suggest to refer in more detail to these discussions. It is necessary that the authors reflect on these already published discussions. Generally, the paper is too simplistic in its reducing the story to aromatic compounds. The authors provide only a selected view on the pathways how organic matter enters the soils. The decomposition of roots is mentioned (although recent references on root biomarkers in soils are missing). However, the input of OM by rhizodeposition is completely ignored. Similarly, the authors consider only sorption of low molecular weight compounds to the solid phase as a mechanism for stabilization of OM in soils. Here again, recent concepts are ignored, e.g. the association of microbial cell wall envelope fragments (see work by Miltner and coworkers). It is necessary to point out that the view of the authors is mainly restricted to acid forest soils, whereas there is also stabilization in neutral pH forest and arable soils where the interaction of acidic compounds with Fe(hydr)oxide surfaces is of minor importance (Kleber et al., 2015). Even if one agrees with all the problems raised, the authors do not provide novel solutions. Solution one is that "careful data interpretation, including critical assessment of experimental and analytical limitations, must become standard". This is a prerequisite of any scientific work and does not tackle the specific problem. The second solution is to use "combined studies on DOM and SOM". Here I agree and I encourage the authors to start such investigations. The figure is just terrible; colors are almost not distinguishable form background. Here a more professional graphics approach is needed to improve the figure.

---

## Referee Comment (RC2) · X. Feng (Referee) · 12 Feb 2016

This paper focuses on the stability of aromatic litter components in the (mineral) soils and provides some interesting insights on the paradoxical evidence of persistence or lability of aromatic organic carbon (OC; including lignin) in the soil and dissolved organic matter (SOM and DOM). This research topic has received increasing attention in the past decades and this opinion paper is timely for synthesizing the mounting (controversial) evidence for/against the stability of aromatic OC in soils (although I feel that more literatures can be included). In addressing this issue, the authors have put a special emphasis on the analytical limitations of detecting and quantifying mineral-protected aromatic OC, which in my opinion is indeed a key bottleneck limiting our understanding on the fate of plant OC in the soils. I therefore wish that the authors may extend the discussions of future research strategies in the last part to provide some "practical"
[Figure]

suggestions on potential research directions or tools to overcome the current analytical weaknesses. For instance, a key issue with the lignin CuO oxidation method is its unknown extract efficiency, which may vary for samples with different mineral matrix or SOM interactions. But this is rarely tested or stated in the papers because we are short-handed dealing with complex macromolecules such as lignin—there is hardly any pure natural macromolecular lignin standard for us to test the method. Can we circumvent this problem using pyrolysis GC/MS or isotopic labeling? Or a combination of different methods may provide further insights on the "hidden" lignin? Before any major analytical breakthrough is made, we must make full use of the current tools for soil lignin studies rather than abandon them, right?

I also agree with the authors that there is a big gap between the long-term fate of lignin and short-term laboratory experiment including sorption studies, in which "hot moments" of lignin transformation may not be captured. For instance, sorption experiments that observed selective binding of aromatics to minerals are typically void of microbial interactions due to the use of HgCl2 (and alike). In natural soils, aromatics that are presumed to sorb selectively to minerals may be preferentially degraded by microbial communities living at the mineral surfaces as well, hence leading to the paradox of "dissolved" and "solid phase line of evidence". Inclusion of both natural geochemical and microbial processes in laboratory or field experiments may be key to finding the missing piece of the soil lignin "jigsaw puzzle"—I believe this goes beyond mineral protection and analytical limitations.

Last but not least, as indicated by the title, the discussion of this paper is focused on plant derived aromatics (lignin and tannin). But how important is black carbon in the overall distribution of aromatic signals in the soil and DOM? Can we distinguish the two? I think it may be useful to briefly differentiate and clarify the fate of lignin/tannin versus black carbon in the soil for the readers' benefit.

Other minor comments: Pg 3, L13: I'm not sure if this sentence is totally true—forest floor is the result of greater litter inputs versus losses through mineralization and

translocation via leaching and bioturbation, etc. For tropical forests with strong litter mineralization, forest floor can be thin or almost absent as well. Actually, I don't think it is necessary to mention the cause for forest floor formation here so this first sentence may be deleted.

Pg 3, L15: decreases with depth.

Pg 3, L20: 25-89% is a very high estimate—does this only apply to soils with limited bioturbation and to subsoils? What about root input? Does this DOC flux include root exudates, which should be differentiated from that leaching from aboveground litter? As root biomass decreases with depths, it is natural that root-derived DOC decreases in mineral soils.

---

## Author Response (AR1)

**Reviewer 1 comment:**

**Reviewer: I generally acknowledge that the authors try to pinpoint and discuss the apparent discrepancies between the fact that large amounts of aromatic compounds are entering the soil in dissolved phase and the fact that they are not found any more sorbed to the solid phase. The authors attribute this to the problems of analyzing lignin and claim that two methods (although completely independent and different in their analytical procedures, i.e. solid-state 13C NMR spectroscopy and CuO oxidation) fail to identify these aromatic compounds. I suggest to carefully check the literature for solid-state 13C NMR work that shows significant contributions of aromatic compounds (although mostly attributed to charred OM, see work by Knicker and coworkers, Skjemstad and coworkers). The authors need to explain why the technique fails to work for lignin-derived aromatic compounds, but does work well for other aromatic (and even more condensed) structures produced from fire impact.**

*Authors:  We thank the reviewer for the critical and constructive comments. The comments greatly helped to increase the quality of the manuscript.*

*We extended the discussion on limitations of analytical methods and refer to the 13C NMR work suggested by the reviewer (see page 8 lines 14 ff.). We agree that this part of the discussion needs to be more precise. The low sensitivity for aromatic compounds is a problem related to CPMAS $^{13}$C-NMR spectroscopy. The Bloch decay technique does not have these limitations, and thus, is more sensitive. This technique is frequently used in studies that specifically address questions on pyrolytic OM. It is, however, not routinely applied in SOM research, probably as longer instrument times are required. The dominance of O/N alkyl C in CPMAS $^{13}$C-NMR of SOM is commonly used as one argument supporting the view that SOM is dominated by microbial-derived compounds. It fits to a number of other observations, and thus we do not doubt that microbial-derived compounds are important contributors to SOM. However, it is seldom discussed that aromatic C might be underestimated by CPMAS $^{13}$C-NMR. Another uncertainty of NMR is related to sample pre-treatment, i.e., mineral soil samples are commonly de-mineralized with hydrofluoric acid in order to remove paramagnetic minerals that disturb the analysis. The treatment can result in significant losses of SOM. In particular, compounds directly bound to the mineral may be preferentially lost. So, we keep to the argument that the interpretation of CPMAS $^{13}$C-NMR literature might add to an underestimation of the turnover rates of plant litter-derived aromatics in soil.*

**Reviewer: The problem that all compounds entering the soil after some decades leave the analytical window for and cannot be identified any more as specific plant or microbial derived compounds has been described and discussed previously and is not specific for aromatic compounds (see detailed discussion of the problem in Hedges et al. (2000). It is also illustrated by the fact that all molecularly identified organic compounds in soil are younger than the mean age of SOM or their turnover is faster than that of bulk SOM (see Amelung et al., Adv in Agronomy, 2008 and later Schmidt et al., Nature, 2011). Thus I suggest to refer in more detail to these discussions. It is necessary that the authors reflect on these already published discussions. Generally, the paper is too simplistic in its reducing the story to aromatic compounds.**

*Authors: We would like to keep focused on plant-derived aromatics, because the contradictions between research on DOM fluxes and research on solid-phase SOM composition are particularly evident for this compounds class. Nevertheless, this is an important comment by the reviewer. Also turnover times of other non-aromatic biomarker compounds might be underestimated because an unknown portion is not extractable (Amelung et al. 2008). We refer to aspects discussed in Amelung et al. (2008) in the revised manuscript (page 7, lines 30 ff.). The discussion now includes a more detailed discussion on possible reasons why only part of the aromatics can be extracted from soil by the CuO method and pyrolysis techniques.*

**Reviewer: The authors provide only a selected view on the pathways how organic matter enters the soils. The decomposition of roots is mentioned (although recent references on root biomarkers in soils are missing). However, the input of OM by rhizodeposition is completely ignored. Similarly, the authors consider**

**only sorption of low molecular weight compounds to the solid phase as a mechanism for stabilization of OM in soils. Here again, recent concepts are ignored, e.g. the association of microbial cell wall envelope fragments (see work by Miltner and coworkers).**

*Authors: We added the suggested aspects to the revised manuscript. The discussion on the possible role of root input as source of SOM was extended (page 5, lines 17 ff.). The possible role of rhizodeposition in SOM turnover is discussed on page 10, lines 4 ff. We also discuss the view of Miltner et al. that some of the organic matter `patches` on clay particles are composed of cell wall structures of microorganisms (page 10, lines 24 ff.). The addition of these aspects greatly improved our discussion about whether limits in process understanding might be the reason for the controversies outlined in the manuscript (chapter 4.2).*

**Reviewer: It is necessary to point out that the view of the authors is mainly restricted to acid forest soils, whereas there is also stabilization in neutral pH forest and arable soils where the interaction of acidic compounds with Fe(hydr)oxide surfaces is of minor importance (Kleber et al., 2015).**

*Authors: This aspect is pointed out in chapter 2.1 on the `dissolved phase line of evidence` (page 4, lines 2 ff.).*

**Reviewer: Even if one agrees with all the problems raised, the authors do not provide novel solutions. Solution one is that "careful data interpretation, including critical assessment of experimental and analytical limitations, must become standard". This is a prerequisite of any scientific work and does not tackle the specific problem. The second solution is to use "combined studies on DOM and SOM". Here I agree and I encourage the authors to start such investigations.**

*Authors: The last part of the manuscript has been revised completely. We summarize key research questions that are still open and provide more distinct suggestions about future strategies (including experimental and modeling work).*

**Reviewer: The figure is just terrible; colors are almost not distinguishable form background. Here a more professional graphics approach is needed to improve the figure.**

*Authors: The figure has been revised.*

**Reviewer 2 comments**

**Reviewer: This paper focuses on the stability of aromatic litter components in the (mineral) soils and provides some interesting insights on the paradoxical evidence of persistence or lability of aromatic organic carbon (OC; including lignin) in the soil and dissolved organic matter (SOM and DOM). This research topic has received increasing attention in the past decades and this opinion paper is timely for synthesizing the mounting (controversial) evidence for/against the stability of aromatic OC in soils (although I feel that more literatures can be included). In addressing this issue, the authors have put a special emphasis on the analytical limitations of detecting and quantifying mineral-protected aromatic OC, which in my opinion is indeed a key bottleneck limiting our understanding on the fate of plant OC in the soils. I therefore wish that the authors may extend the discussions of future research strategies in the last part to provide some "practical" suggestions on potential research directions or tools to overcome the current analytical weaknesses. For instance, a key issue with the lignin CuO oxidation method is its unknown extract efficiency, which may vary for samples with different mineral matrix or SOM interactions. But this is rarely tested or stated in the papers because we are short-handed dealing with complex macromolecules such as lignin. There is hardly any pure natural macromolecular lignin standard for us to test the method. Can we circumvent this problem using pyrolysis GC/MS or isotopic labeling? Or a combination of different methods may provide further insights on the "hidden" lignin? Before any major analytical breakthrough is made, we must make full use of the current tools for soil lignin studies rather than abandon them, right?**

*Authors: Dear Dr. Feng, we thank you for the effort that has gone into evaluating our article. Your suggestions and comments led to a greatly improved manuscript. We agree that the part on future research strategies was too short and vague. The problems of quantifying total amounts of aromatics in soil may not be fully solvable with currently available analytical methods. However, strategies of combining different methods could help to gain improved estimates on how much aromatics might be `hidden` in soil. We added a more comprehensive discussion on the problem that not all of the aromatics can be extracted from soil (page 7, lines 27 ff.). In addition, the last part of the manuscript has been entirely revised and extended. It now presents a list of possible future research strategies that might help to step forward on solving the problems outlined in our article. The list includes open research questions as well as suggestions on methodological approaches.*

**Reviewer: I also agree with the authors that there is a big gap between the long-term fate of lignin and short-term laboratory experiment including sorption studies, in which "hot moments" of lignin transformation may not be captured. For instance, sorption experiments that observed selective binding of aromatics to minerals are typically void of microbial interactions due to the use of HgCl2 (and alike). In natural soils, aromatics that are presumed to sorb selectively to minerals may be preferentially degraded by microbial communities living at the mineral surfaces as well, hence leading to the paradox of "dissolved" and "solid phase line of evidence". Inclusion of both natural geochemical and microbial processes in laboratory or field experiments may be key to finding the missing piece of the soil lignin jigsaw puzzle". I believe this goes beyond mineral protection and analytical limitations.**

*Authors: We extended the discussion on current limits of process understanding (see chapter 4.2). One major aspect of the discussion is that the `fate` of sorbed material is hardly understood. We agree, the view that DOM leached from decomposing litter sorbs onto mineral surfaces and then contributes to stable SOM might be too simplistic. Further transformations, desorption, and transport of the sorbed OM might occur in soil. Possibly this depletes SOM in aromatics.*

**Reviewer: Last but not least, as indicated by the title, the discussion of this paper is focused on plant derived aromatics (lignin and tannin). But how important is black carbon in the overall distribution of aromatic signals in the soil and DOM? Can we distinguish the two? I think it may be useful to briefly differentiate and clarify the fate of lignin/tannin versus black carbon in the soil for the readers' benefit.**

*Authors: We decided to keep focused on plant litter-derived aromatics, but added a discussion on possible contribution of pyrogenic OM (page 8, lines 30 ff.). Current available literature on DOM mainly considers plant litter-derived aromatic compounds. Numerous studies addressed in the production of aromatics in forest floors during litter decomposition. Few studies are available on role of pyrogenic OM for DOM fluxes. Many studies on DOM use indicators for bulk aromatics (e.g., UV absobance). Hence, the unknown contribution of pyrogenic OM to DOM is an uncertainty in the interpretation of DOM flux data.*

**Reviewer: Other minor comments: Pg 3, L13: I'm not sure if this sentence is totally true forest floor is the result of greater litter inputs versus losses through mineralization and translocation via leaching and bioturbation, etc. For tropical forests with strong litter mineralization, forest floor can be thin or almost absent as well. Actually, I don't think it is necessary to mention the cause for forest floor formation here so this first sentence may be deleted.**

*Authors: We shortened the paragraph and deleted the discussion on bioturbation. We agree, it is a side aspect and not necessary to mention.*

**Reviewer: Pg 3, L15: decreases with depth.**

*Authors: Sentence has been removed.*

**Reviewer: Pg 3, L20: 25-89% is a very high estimate does this only apply to soils with limited bioturbation and to subsoils? What about root input? Does this DOC flux include root exudates, which should be differentiated from that leaching from aboveground litter? As root biomass decreases with depths, it is natural that root-derived DOC decreases in mineral soils.**

*Authors: We present more details on the work cited in this paragraph. The estimates are for acidic forest soils with permanent forest floor. Hence, there is indeed limited bioturbation. The model calculations base on measured DOC fluxes, DOM sorption and mineralization. The role of roots as source of DOM is not clear. In general, there is little information available on that issue. We discuss it in the revised manuscript (page 5, lines 20 ff.).*

**Gone or just out of sight? The apparent disappearance of aromatic litter components in soils**

**T. Klotzbücher[1], K. Kalbitz[2], C. Cerli[3], P.J. Hernes[4], K. Kaiser[1]**

[1] Soil Science and Soil Protection, Martin Luther University Halle-Wittenberg, von-Seckendoff-Platz 3, 06120 Halle (Saale), Germany

[2] Institute of Soil Science and Site Ecology, Technical University Dresden, Pienner Strasse 19, 01737 Tharandt, Germany

[3] Institute of Biodiversity and Ecosystem Dynamics, Earth Surface Science, University of Amsterdam, POSTBUS 94240, 1090 GE Amsterdam, The Netherlands

[4] Department of Land, Air, and Water Resources, University of California, One Shields Avenue, Davis, California 95616, United States

Correspondence to: T. Klotzbücher (thimo.klotzbuecher@landw.uni-halle.de)

**Abstract**

Uncertainties concerning stabilization of organic compounds in soil limit our basic understanding on soil organic matter (SOM) formation and our ability to model and manage effects of global change on SOM stocks. One controversially debated aspect is the contribution of aromatic litter components, such as lignin and tannins, to stable SOM forms. In the present opinion paper, we summarize and discuss the inconsistencies and propose research options to clear them.

Lignin degradation takes place step-wise, starting with (i) depolymerisation, followed by (ii) transformation of the water-soluble depolymerization products. The long-term fate of the depolymerization products and other soluble aromatics, e.g., tannins, in the mineral soils is still a mystery. Research on dissolved organic matter (DOM) composition and fluxes indicates dissolved aromatics are important precursors of stable SOM attached to mineral surfaces and persist in soils for centuries to millennia. Evidence comes from flux analyses in soil profiles, biodegradation assays, and sorption experiments. In contrast, studies on composition of mineral-associated SOM indicate the prevalence of non-aromatic microbialderived compounds. Other studies suggest the turnover of lignin in soil can be faster than the turnover of bulk SOM. Mechanisms that can explain the apparent fast disappearance of lignin in mineral soils are, however, not yet identified.

The contradictions might be explained by analytical problems. Commonly used methods probably detect only a fraction of the aromatics stored in the mineral soil. Careful data interpretation, critical assessment of analytical limitations, and combined studies on DOM and solid-phase SOM could thus be ways to unveil the issues.

**1   Introduction**

Storage and quality of soil organic matter (SOM) determine many crucial soil properties and the cycles of carbon (C) and essential nutrients through ecosystems. The storage of SOM is determined by plant litter inputs and decomposition processes. Decomposition of SOM is a significant source of atmospheric $CO_2$, thus, a critical parameter in climate models (Schlesinger and Andrews, 2000). Decomposition rates are sensitive to global change factors such as temperature, precipitation, and land use. However, our ability to understand and predict such responses is limited by uncertainties about pathways of organic matter transformation in soil. In particular, the question as to why some SOM components persist in soil for centuries (denoted as `stable SOM` from here on) while others turn over quickly is still puzzling (Schmidt et al., 2011).

Recent research challenges traditional theories presuming that stable SOM results from neoformation of complex humic polymers in soil (`humification`). Stable SOM rather seems to be composed of relatively simple organic compounds that are protected against biodegradation, e.g., because they are tightly bound to mineral surfaces (Schmidt et al., 2011; Kleber et al., 2015). Herein, we hold to this view but argue that, despite extensive research in the last years, the chemistry and source of compounds incorporated into stable SOM is still largely uncertain. In particular, the importance of aromatic compounds  derivied from abundant plant litter components, such as lignin and tannins, is controversially debated (Figure 1). One line of evidence suggests that they are important contributors to stable SOM. It bases primarily on data from research on fluxes and behaviour of dissolved organic matter (DOM) in soil, hence, we will denote it as the `dissolved phase line of evidence`. A contrasting line of evidence  suggests a quick degradation of aromatic compounds in soil derives primarily from analyses  of the

composition of solid SOM (-`solid phase line of evidence`). Herein, we sum up and confront the arguments of the two views, discuss potential reasons for the controversies (including limitations in analytical methods and process understanding) as well as their implications for our basic understanding on SOM formation.

**2    Dissolved phase line of evidence**

The view that plant-derived aromatics are a major source of stable SOM bases is based on the following two main arguments:

(1) DOM produced during litter decomposition and leached into mineral soil is a main source of stable SOM adsorbed on mineral surfaces.

(2) Aromatic DOM components produced during litter decomposition are resistant against being mineralizedto mineralization and preferentially sorb to mineral surfaces. Hence, they are preferentially stabilized in mineral soil.

**2.1    Argument 1: DOM as source of stable SOM**

Leaching of DOM is a major pathway for organic matter translocation from forest floor into the topsoil horizons. Dissolved OM represents only a small fraction of total SOM. However, it is continuously produced and transported within profiles. In forest soils with permanent forest floor horizons, much of the translocation of organic matter from the forest floor into mineral topsoils is due to DOM leaching, while bioturbation plays either a minor role or no role at all. For such systems, it has been estimated that 10-25% of the total C input to the forest floor via litter fall is leached into mineral soils in the form of dissolved organic C (DOC) (Guggenberger and Kaiser, 2003). 
[revised manuscript text omitted]

[Figure]

[Figure]

1 .

---

## Author Response (AR2)

1  **Dear Dr. Rumpel, we thank you for the effort that has gone into evaluating our article.**

2  **Your suggestions greatly helped to improve the quality of the manuscript. Please find**

3  **below our replies to the individual comments.**

5  • The manuscript should be carefully proofread, e.g. in the first sentence it should be
6  '….determine many crucial soil propertis and the cycling of carbon ….' ; l.26
7  'deri*ied' should be 'derived' ; P.2, l. 4 : 'understanding of SOM formation' ; P.6, l.
8  27 : 'are relatively small components' should be replaced by 'contributing little to', P.
9  8, l.7 : 'aromatic matter' should be replaced by 'aromatic OM compounds' ; P.12, l.11
10  : 'on' should be deleted.

12  **Answer: We proofread the manuscript and corrected these sentences.**

14  • In the two sections giving the arguments for contradictory theories for stabilisation of
15  aromatic C, authors should discuss all literature in terms of positive arguments for
16  these research ligne.

18  **Answer: At first, we planned to write a comprehensive literature review on the topic,**

19  **but then decided to write the manuscript in a more condensed manner, because for**

20  **many of the individual arguments that we discuss, review articles have been published**

21  **during the last years.**

22  **For instance, Kleber et al. (2015) recently presented a comprehensive literature review**

23  **and discussion on importance of sorption of DOM to mineral surfaces for the long-term**

24  **preservation of organic matter in soil. Thevenot et al. (2010) summed up the large body**

25  **of literature (about 150 articles) on distribution of lignin phenols in mineral soils. In the**

26  **revised manuscript, we now clearly indicate the cases, in which we refer to data and**

27  **conclusions discussed in review articles.**

28  **In other cases, however, when no recent comprehensive review articles are available, we**

29  **now present a list of relevant studies supporting the argumentation. This resulted in the**

30  **addition of about 40 references to the manuscript.**

32  • In the 'Dissolved phase line of evidence', the third paragraph on the importance of
33  root origin of dissolved aromatics should be the third argument. In this paragraph it

should be stated that all these arguments were obtained for acid forest soils under temperate climate or through laboratory analyses. Therefore, this line of evidence is difficult to use as a general argument.

**Answer: Done. We now present the paragraph as the third argument supporting the `dissolved phase line of evidence`. In addition, we outline that the currently available data on release of aromatic DOM components is limited to laboratory experiments and refers to root decomposition in acid temperate forest soils. We agree, knowledge about how root decomposition contributes to DOM in mineral soil is currently very limited.**

- In the second section 'solid line of evidence', the contradictory argument mentioned in the last sentence (P.7, l.30-32) should be deleted, as it introduces a confusion to the reader.

**Answer: Done.**

- P. 8, l.29: the last sentence should be deleted, as it is a general statement. Instead it could be good to add a transition sentence introducing NMR as a method to quantify aromatic compounds.

**Answer: Done. The sentence ` Our knowledge about how much lignin is `hidden` (Hernes et al., 2013) in soil is still insufficient` has been deleted. In addition, we revised the first sentence of the paragraph on NMR. It now reads: ` Solid-state CPMAS $^{13}$C-NMR has been widely used in the last decades to study the composition of SOM. Whether the results are quantitative has been subject of an intensive debate`.**

- P.9 : Concerning the analytical bias introduced by HF treatment, the study by Eusterhues et al. (2007) , OG should be considered. It shows that the composition of HF soluble C is different for different soil types – thus not providing evidence of selective loss of specific C species.

**Answer: We discuss the article of Eusterhues in the revised manuscript. In our opinion, the comparison of NMR spectra from untreated and HF-treated soil comes with uncertainties**. In the untreated soil, part **of the SOM attached to mineral surfaces might**

have been invisible due to the proximity to paramagnetic material, while in the HF-treated soil, the same SOM might have become removed during the treatment, thus, was no more detectable. Moreover, the data presented in Eusterhues et al. (2007) suggest that not all of the minerals were removed by the HF-treatment, presumably as the minerals were protected by OM coatings. In topsoil, only 20-30% of the mineral-associated OC was removed. Hence, it was not possible to gain any information about the composition of the other 70-80%.

- P.9 : the Bloch decay technique may lead in some cases to underestimation of O-akyl C, whereas the work of Knicker shows nicely that aromatic C, if not very condensed can be detected by CPMAS.

**Answer: We agree that aromatic C can be detected in soil with the CPMAS method. However, it is likely that its contribution is underestimated by the common approach of integrating and comparing peak areas of different regions of the spectra. Evidence for this can be found in literature. We added a more comprehensive discussion on this issue to the revised manuscript. It reads as follows: "The common approach used to quantify the relative contribution of different C types to SOM is to integrate and compare peak areas of different spectra regions without considering any non-proportional signal responses (e.g., Kögel-Knabner 2002). It has been shown that this approach underestimates lignin vs. cellulose in ligno-cellulose isolated from wheat (Gauthier et al., 2002). Also, peaks of methoxyl C often tend to be larger than those of aromatic C. Both, methoxyl C and aromatic C derive from lignin, but lignin has much less methoxyl C than aromatic C. Spectra of wood samples (e.g., Bonanomi et al. 2014), needle litter (Preston and Trofymow, 2015) and grass litter (McKee et al., 2016) all show that patterns of methoxyl signals being larger than those of aromatic and phenolic C combined."**

**Moreover, we added a sentence addressing problems related to the Bloch decay technique. Our literature review showed that the commonly mentioned problem is the generally low signal intensity.**

- The discussion about pyrogenic C should be moved to the process understanding part. The last sentence (P.9, l29-31) can be deleted.

**Answer: Done.**

- P.10, l.1 : a sentence is needed introducing the fact that process understanding may be limited by several different aspects related to our conceptual and experimental approaches.

**Answer: We added the following introductory sentence: "The contradictions outlined herein might also suggest gaps in the understanding of SOM turnover processes. Here we argue that, in particular, knowledge about the turnover of SOM at mineral surfaces is insufficient. This is due to the yet uncertain quantitative composition of SOM. In addition, prevailing conceptual ideas and paradigms have been questioned in recent years."**

- 4.2. Limits in process understanding : I think that this paragraph need to be better structured – temporal aspect as discussed in the beginning should be separated from spatial ones (probalby the last paragraph could be moved up a little); plant activity effects should be separated from the processes depending on pedology

**Answer: We revised the first part of the chapter on process understanding in order to improve the structure. First, we present an introductory paragraph (see above), which argues that uncertainties about processing of sorbed OM represent a major knowledge gap. We then discuss current findings and hypotheses about processing of sorbed OM (i.e., the `cycling downwards` model of Kaiser and Kalbitz and the possible role of root exudates for turnover of sorbed SOM). Thereafter we discuss recent findings on the nm-scale distribution of SOM at mineral surfaces, which might have implications for the understanding of the processes occurring at mineral surfaces.**

- P.11, l.31 : 'and methodological problems ' should be added after 'the contradictions'

**Answer: Done.**

- P.12/13 : in the research strategies it shold be mentioned that the origin of aromatics in water and soil should be elucidated and specific stabilisation and destabilisation mechanisms studied (e.g. photooxidation ; binding of tannins to proteins….)

**Answer: We mention the problem that the origin of the aromatic DOM is oftentimes uncertain in the first point of the research strategies. Moreover, we added the following paragraph to the research strategies (second point): "
[revised manuscript text omitted]

---

## Author Response (AR3)

Dear Dr. Rumpel,

Again, thank you for your suggestions and for the productive discussion on the manuscript throughout the review process. The two suggested revisions on the latest manuscript version have been done: (i) we cite Lehmann and Kleber (2015) on P.2, line 21, and (ii) we deleted the paragraph of P.9, lines 20-24. We did not change the title because we think that the question on fate of plant-derived aromatic compounds in mineral soil is still the focal topic of the manuscript.

Best regards,

Thimo Klotzbücher

[revised manuscript text omitted]